# Expression of Nerve Growth Factor and Its Receptor TrkA in the Reproductive System of Adult Zebrafish

**DOI:** 10.3390/vetsci9050225

**Published:** 2022-05-06

**Authors:** Pietro Cacialli

**Affiliations:** Department of Veterinary Medicine, University of Naples Federico II, 80138 Napoli, Italy; pietro.cacialli@unige.ch

**Keywords:** zebrafish, NGF, TrkA, ovary, testis

## Abstract

Nerve growth factor (NGF), a member of the neurotrophin family, has emerged as an active mediator in different crucial events in the peripheral and central nervous system. At the same time, several studies showed that this neurotrophin can also play a role in non-neuronal tissues (e.g., among gonads). In spite of a large number of studies present in mammals, investigations devoted to NGF and its receptor TrkA in the reproductive system of other animal models, such as teleost fish, are scarce. To increase our knowledge of NGF and its receptor in a vertebrate gonads model, the present report describes the expression patterns of *ngf* and *trka* mRNA in the testis and ovary of adult zebrafish. By using chromogenic and fluorescence in situ hybridization, we demonstrate that in the testis of adult zebrafish, *ngf* and its receptor *trka* are mainly expressed in spermatogony B and spermatocytes. In the ovary of this fish, *ngf* and *trka* are expressed at different stages of oocyte development. Altogether, these results show that this neurotrophin and its receptor have an important role in the reproductive system that is conserved during vertebrate evolution.

## 1. Introduction

Nerve growth factor (NGF) has been the first discovered member of the neurotrophin family [1,2]. Additional components of this group are brain derived neurotrophic factor (BDNF); neurotrophin 3 (NT3); neurotrophin 4/5 (NT4); and neurotrophin 6/7 (NT6/7) [3,4,5,6]. All neurotrophins can be synthesizes as precursors. Indeed, NGF also presents a precursor form (pro-NGF) and is either secreted outside the cells, as pro- and mature NGF, or is cleaved intracellularly to mature NGF. Both forms are active [7] and homodimers [8]. The *ngf* gene, as with other neurotrophins, is well conserved across vertebrate evolution, suggesting conserved functions [9,10,11,12,13]. It can bind a tyrosine kinase receptor TrkA and/or p75 neurotrophin receptor [14,15]. The binding with TrkA receptor is essential to ensure the survival of neurons and induce their growth in the peripheral and central nervous systems [16]. Otherwise, binding with p75 induces cell death. In vertebrates, this neurotrophin and TrkA receptor are also important in peripheral tissues [17,18,19,20,21,22], as reported by their presence in several organs (such as the reproductive system) [23,24]. Previous studies from human, rat, mice, wild ground squirrel, and Japanese quail have shown that this neurotrophin and TrkA receptor can fill a crucial function in the integrity of ovarian sympathetic innervation [25,26,27,28,29]. In these species, NGF and its receptor have been detected in different populations of ovary constituent cells, such as oocytes; interstitial cells within the ovary; theca; and granulosa cells [30,31,32]. It has been widely reported that NGF/TRKA signaling pathway is required to facilitate the ovulatory process and ensure the development of primordial ovarian follicles [33]. In non-mammalian vertebrates, NGF and its receptor have been found in Xenopus ovaries. Further functional studies reported that NGF treatment induced meiotic maturation of Xenopus oocytes overexpressing TrkA receptor [34]. At the same time, it has been shown that this signaling pathway appears to be involved in spermatogenesis [35]. In the testes of several mammal species, NGF has been observed in elongated spermatids, pachytene, and primary spermatocytes [36,37,38]. Differently, its receptor TrkA is present only in elongated spermatids and membranes of Leydig cells. Concerning the functional role during spermatogenesis [39] and testicular development, recent studies have shown that NGF and its receptor play an important role promoting testosterone production, proliferation, and differentiation of Leydig cells [40]. Low levels of *ngf* mRNA in the sperm have been associated with azoospermia in human patients. These results have been confirmed by functional investigations in rabbit and mice [41], in which the inhibition of the NGF/TrKA signaling pathway induced a specific decrease of sperm motility. More recent studies have shown that the administration of this neurotrophin can be used as a potential therapy to restore spermatogenesis in a new mice model affected by severe testicular atrophy of the seminiferous tubules [42]. In spite of a large number of studies present in mammals, investigations concerning this neurotrophin and its specific receptor in the reproductive system of other animal models, such as fish, are scarce. In the present study, we report for the first time the expression patterns of *ngf* and *trka* in the reproductive system of *Danio rerio* (i.e., zebrafish). This fish presents different features (small size, rapid fertilization, and fast embryonic development) making it an excellent animal model for biomedical and veterinary research. This fascinating model can be utilized in mutagenesis analyses and drug screening tests. It has been used to conduct experiments related to different research fields such as embryology, genetics, cancer, cardiovascular, and organ and tissue regeneration [43,44,45,46]. In the present report, firstly, in order to identify different cell populations in adult zebrafish testis and ovary, hematoxylin-eosin staining on paraffin sections has been carried out. Next, by using chromogenic and fluorescence in situ hybridization, it has been shown that *ngf* and *trka* mRNA were highly expressed during spermatogenesis and oogenesis of adult zebrafish, confirming their role during vertebrate evolution.

## 2. Material and Methods

### 2.1. Animals and Gonads Dissection

All experiments have been performed according to the Italian Decree 26/2014, and approved (n°2/2020-PR), by the Committee of the University of Naples Federico II. Adult females and males (1 year) of Danio rerio were anesthetized using 0.3% aminobenzoic acid-ethyl-methylester (MS222, Sigma, St. Louis, MO, USA). The ovary and testis were dissected and fixed for one day in 4% paraformaldeyde, after dehydration, included in paraffin and then sectioned. The sections were used for hematoxylin-eosin staining and in situ hybridization (for detail see Section 2.2 and Section 2.3).

### 2.2. Histology (Haematoxylin and Eosin Staining)

Ovary and testis were fixed in 4% paraformaldehyde in PBS for 24 h at 4 °C. Next, the tissues were washed and stored in ethanol (70%) at 4 °C. After, ovary and testis were embedded in paraffin and sectioned using a rotary microtome. The paraffin section were mounted on slides. To deparaffinize, all paraffin sections were immersed in xylene, and rehydrated in ethanol 100%, 95%, 80%, 70%, and 50%. Next, the paraffin sections were washed three times in distilled H_2_O (1 min). Based on the morphological features described by previous studies [47,48], to examine cell and tissue morphology of ovary and testis, it has been used hematoxylin and eosin (HE) staining.

### 2.3. Chromogenic and Fluorescence In Situ Hybridization (ISH)

Digoxigenin (DIG)-labeled antisense riboprobes, were prepared as reported in previous studies [49,50,51]. In detail, *ngf* and *trka* expressions were detected using in situ hybridization (ISH). To generate *ngf* and *trka* probes, have been linearized the vectors ZeroBlunt and or TOPO-TA holding the product amplified using the polymerase chain reaction. Next to produce antisense and sense ribo-probes, the plasmids were linearized with the specific enzymes. After plasmid linearization, it has been performed in vitro transcription, using SP6 and or T7 polymerase (both from Roche-Diagnostic, Chicago, IL, USA), adding the label mix digoxigenin RNA. Next, to validate the specificity of the riboprobes, the anti and sense probes were hybridized on adjacent sections (see Appendix A). To generate *ngf* and *trka* riboprobes, have been used the following primers:
*ngf* F: GGAGCACAGGAGATCTACGC and R: CGTGGAAAAACCCAACTCAT;*trka* F: AGTTGTTGCTTGCAGGGTGG and R: TGGGTCAATCATGACCTCAG.

After the production of the specific riboprobes, ovary and testis were quickly dissected and fixed in 4% paraformaldehyde, overnight at 4 °C. After 24 h, they were processed for paraffin. The slides (10 µm) were obtained using a rotary microtome and mounted on slides. To deparaffinize, all paraffin sections were immersed in xylene two times (3 min), and rehydrated in ethanol 100%, 95%, 80%, 70%, 50% (3 min each). To fix, the sections were immersed for 20 min in 4% paraformaldehyde. Then, the sections of ovary and testis immersed in PBS and adding the proteinase K diluted 2 mg/mL at room temperature for 7 min. Next, all slides were processed as follows: fixed in 4% paraformaldehyde for 20 min, washed in PBS and standard saline sodium citrate (SSC 2x) 2 times (10 min each). Next, the slides were incubated at 63 °C for 24 h, using a moist chamber with the probes (2 µg/mL) diluted in a specific medium (Denhart 5x; SSC 2x; 50% formamide; ethylenediamine-tetra acetic acid 4 mM; 5% dextran sulfate; yeast tRNA 50 µg/mL). After 24 h, the section was washed with SCC 2x; 50% formamide/SCC 2x; SSC 0.2x and SSC 0.1x. The sections were immersed in a buffer Tris-HCl/NaCl (mixing 100 mM of Tris-HCl pH 7.5 and 150 mM NaCl) and washed in the same buffer containing 0.5% milk powder and adding 0.1% Triton.

**For chromogenic ISH**, during the second day, all sections have been incubated with anti-digoxigenin alkaline phosphatase Fab fragments, dilution 1:5000 (Roche Diagnostic company, Chicago, IL, USA), overnight at room temperature (RT). After 24 h, all sections were washed in Tris-HCl/NaCl buffer and with 110 mM HCl-Tris (pH 8) containing 10 mM MgCl_2_ and 110 mM NaCl. Staining was performed using NBT/BCIP buffer (pH 9.5).

**For fluorescence ISH,** all slides were immersed in anti-DIG POD antibody (Roche) 1:200 dilution in the above blocking solution at room temperature for 24 h. Next, the slides were washed as follow: 4 times in maleic acid buffer (20 min each); 4 times in PBS (10 min each). Next, the sections were incubated for 1 h in a buffer for amplification (PerkinElmer company, Life Sciences, Boston, USA). For the reaction, it has been prepared Cy3 tyramide (TSA plus Cyanine 3, PerkinElmer company, Life Sciences, Boston, USA) reagent 1:100 in amplification buffer and AlexaFluor 488 reagent (TSA Reagent, Alexa Fluor 488 Tyramide Reagent, Invitrogen™, Boston, MA, USA). Next the slides have been immersed several times (10 min each), mounted with DAPI and coverslide and then observed with a confocal Nikon Eclipse 90i microscope. All pictures have been acquired using a software (NIS-Elements 4.2, Nikon, Milan, Italy).

## 3. Results

### 3.1. Morphological Analysis of Adult Zebrafish Testis

As reported also in previous morphological studies, spermatogenesis in zebrafish occurs in cysts (Figure 1a–c).

These cysts can be formed after several Sertoli cells are positioned around the spermatogonium in the cytoplasm. Numerous cysts can be observed, characterized by spermatogonia A (round shape), presenting a different number of nucleoli (one, two or more); spermatogonia B, presenting a rising presence of heterochromatin in elongated and/or round nuclei (it can also be present small nucleoli); spermatocytes, which can be identified by several steps of meiotic division, observing the size of the nucleus and chromosome condensation; and early, intermediate, and mature spermatids, which present a significant decrease of the cellular and nuclear volumes (Figure 1a–c).

### 3.2. Ngf and Its Receptor trka Are Expressed in Adult Zebrafish Testis

Next, it has been performed chromogenic and fluorescence in situ hybridization, to identify the expression patterns of *ngf* and its receptor *trka,* in adult zebrafish testis.

*Ngf* was mainly expressed in spermatogony B and spermatocytes (dyplotenic phase) (Figure 2a–f). These results have been confirmed by using confocal microscopy with high magnification (Figure 3a,b).

Concerning the receptor *trka,* it was expressed in three different cell populations: spermatogony type A, B and dyplotenic spermatocytes of adult zebrafish testis (Figure 4a–f). These results were confirmed by using confocal microscopy with high magnification (Figure 5a–c).

### 3.3. Morphological Analysis of Zebrafish Ovary

As reported in previous studies, one can identify five different oocyte stages in zebrafish ovaries. (Figure 6a–h).

### 3.4. Ngf and Its Receptor Trka Are Expressed in Adult Zebrafish Ovary

Next, chromogenic and fluorescence in situ hybridization has been used to identify the expression patterns of *ngf* and its receptor *trka* in adult zebrafish ovaries.

*Ngf* was mainly expressed in the perinuclear cytoplasm of oocytes at the stage I, II, and III (Figure 7a,b). At the stage IV and V, *ngf* was expressed in the follicular cells located around the oocyte (Figure 7c,d). The receptor *trka* was more expressed in the perinuclear cytoplasm of oocytes at the stages II and III (Figure 8a,b). As *ngf*, the receptor *trka* was also expressed in follicular cells of oocyte at the stage V (Figure 8c,d).

## 4. Discussion

For the first time, in the present report, the expression patterns of *ngf* and its receptor *trka* in the reproductive system of adult zebrafish have been described. Firstly, it has been shown that *ngf* and *trka* are specifically expressed in spermatogony A and/or B and dyplotenic spermatocytes. Therefore, assuming that they have an important role during the spermatogenesis in the testis of adult zebrafish. Previous observations in rat and mice models reported that NGF was expressed in elongated spermatids, pachytene, and primary spermatocytes. In contrast, its receptor TrkA has been detected only in the membranes of Leydig cells and elongated spermatids [52,53,54]. In addition, recent studies reported that NGF and TrkA can stimulate sperm motility and induce sperm cell acrosome reactions. Furthermore, NGF plays an important role promoting testosterone production, proliferation and differentiation of Leydig cells [55,56]. Similar findings were also reported in human, monkey and rabbit. Differently, in mature alpacas, NGF protein was mainly express in the perinuclear cytoplasm of stromal cells, Sertoli cells and germ cells [57]. On the evidence that NGF and its receptor TrkA have been specifically detected in the germinal and/or endocrine cells of the testis, it has been speculated that NGF can play an autocrine and/or paracrine signaling during testicular development and spermatogenesis. In detail, it has been reported that the synthesis of NGF occurs in somatic cells (Leyding), after this neurotrophin can bind the receptor TrkA expressed in germ cells, such as; spermatocytes and spermatids, regulating the different stages of the spermatogenesis in rat and ground squirrel. This result has been observed also in patients affected by azoospermia, all of whom presented low levels of NGF [58,59,60]. These findings have been confirmed by functional investigations in rabbit and mice [61], in which the inhibition of the NGF/TrKA signaling pathway induced a specific decrease of sperm motility. More recent study also reported that the administration of this neurotrophin can be used as a potential therapy to restore spermatogenesis in a new mice model affected by severe testicular atrophy (and characterized by the loss of germ cells).

With regard to the expression of *ngf* and *trka* in the ovaries of adult zebrafish, the results show that they are expressed in oocytes at different stages. In particular, *ngf* and its receptor are detected in oocytes at stages II–III and in follicular cells of oocytes at stages IV and/or V. Previous studies in different species show that NGF and its receptor play a pivotal role during the differentiation of primordial ovarian follicles and the growth of the latter in secondary follicles. In detail, it has been shown that NGF/TrkA signaling pathway induced the proliferation of theca and granulosa cells, increasing the synthesis of the follicular stimulating hormone (FSH) receptor in granulosa cells. Concerning the ovulation, the NGF/TrkA pathway is highly involved in this process. Indeed, this pathway, in pre-ovulatory follicle, can modulate the release of prostaglandin-E_2_ (PGE_2_), inhibiting the expression of gap-junctions and increasing the proliferation of theca cell [62,63,64]. High levels of NGF were reported in goat and wild ground squirrel ovaries. In human ovary [65], NGF was detected in the oocytes and granulosa cells of preantral follicles, from the primordial to the secondary stages [66]. Further studies showed that human fetal oocytes presented mostly full cytoplasmic staining for NGF, whereas oocytes from adults exhibited mostly partial cytoplasmic and nuclear staining. At the same time, in vitro studies showed that treatment can inhibit the apoptosis by down-regulating apoptotic genes in mouse ovary. The dynamic expression of NGF and its receptor TrkA, have been associated with a luteotropic effect of ovulation-inducing factor in bovine ovary [24]. These results have been confirmed also in other species such as cows and pigs [67]. Most important is that these findings have been also confirmed in non-mammalian vertebrates (such as Xenopus). In this species, NGF and its receptor have been detected in the ovary, during early stages of oocytes maturation [34]. Indeed, functional studies showed that NGF administration induced meiotic maturation of Xenopus oocytes overexpressing TrkA receptor.

## 5. Conclusions

This study, for the first time, provides a precise and comprehensive description of the expression patterns of *ngf* and its receptor *trka* in the reproductive systems of an adult teleost fish, zebrafish. This report showed that *ngf* and its receptor are mainly stored in spermatogonia and spermatocytes in this fish, as reported in previous data from different mammal species. At the same time, this study provides new results concerning *ngf* and *trka,* both are mostly expressed during early stages and in follicular cells (stage V) of oogenesis in fish. Taken altogether these results show that this neurotrophin and its receptor have an important role in the reproductive system which is conserved during vertebrate evolution.

## Figures and Tables

**Figure 1 vetsci-09-00225-f001:**
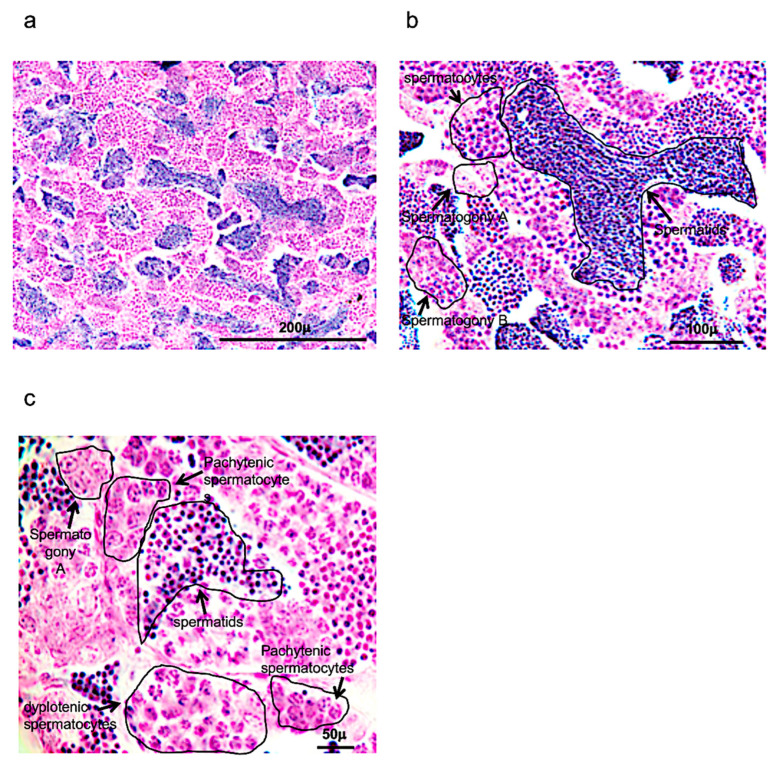
Hematoxylin-eosin staining of adult zebrafish testis. (**a**) Overview of adult zebrafish testis by hematoxylin-eosin staining. (**b**) Spermatogony A-B, spermatocytes and spermatids in adult zebrafish testis by hematoxylin-eosin staining. (**c**) High magnification of spermatogony A, spermatocytes (different phases of meiosis) and spermatids in adult zebrafish testis by hematoxylin-eosin staining. Scale bars are: 200 µ (**a**); 100 µ (**b**); and 50 µ (**c**).

**Figure 2 vetsci-09-00225-f002:**
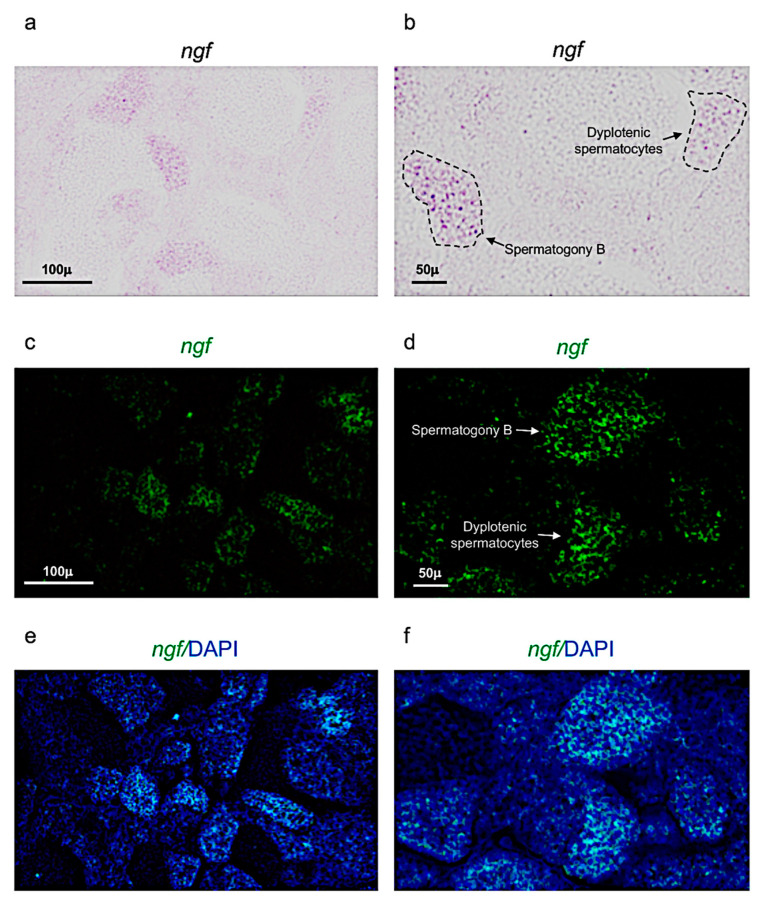
*Ngf* expression pattern in adult zebrafish testis. (**a**,**b**) Chromogenic in situ hybridization of *ngf* in adult zebrafish testis. (**c**,**d**) Fluorescence in situ hybridization of *ngf* in adult zebrafish testis. (**e**,**f**) Fluorescence in situ hybridization of *ngf* and cell nuclei (DAPI), in adult zebrafish testis. Scale bars are: 100 µ (**a**,**c**,**e**); 50 µ (**b**,**d**,**f**).

**Figure 3 vetsci-09-00225-f003:**
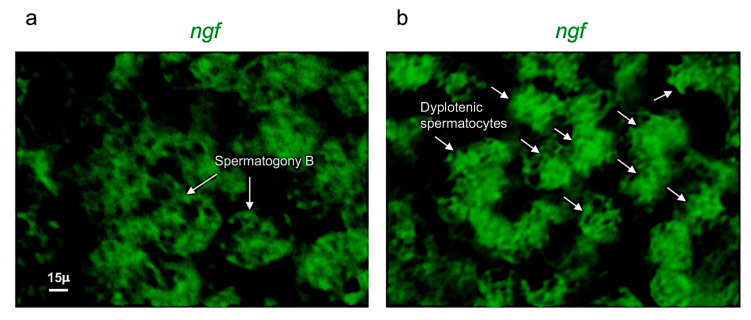
*Ngf* is specifically expressed in spermatogony B and dyplotenic spermatocytes. High magnification, fluorescence in situ hybridization of *ngf*, specifically expressed in (**a**) spermatogony B and in (**b**) dyplotenic spermatocytes in adult zebrafish testis. Scale bar: 15 µ (**a**,**b**).

**Figure 4 vetsci-09-00225-f004:**
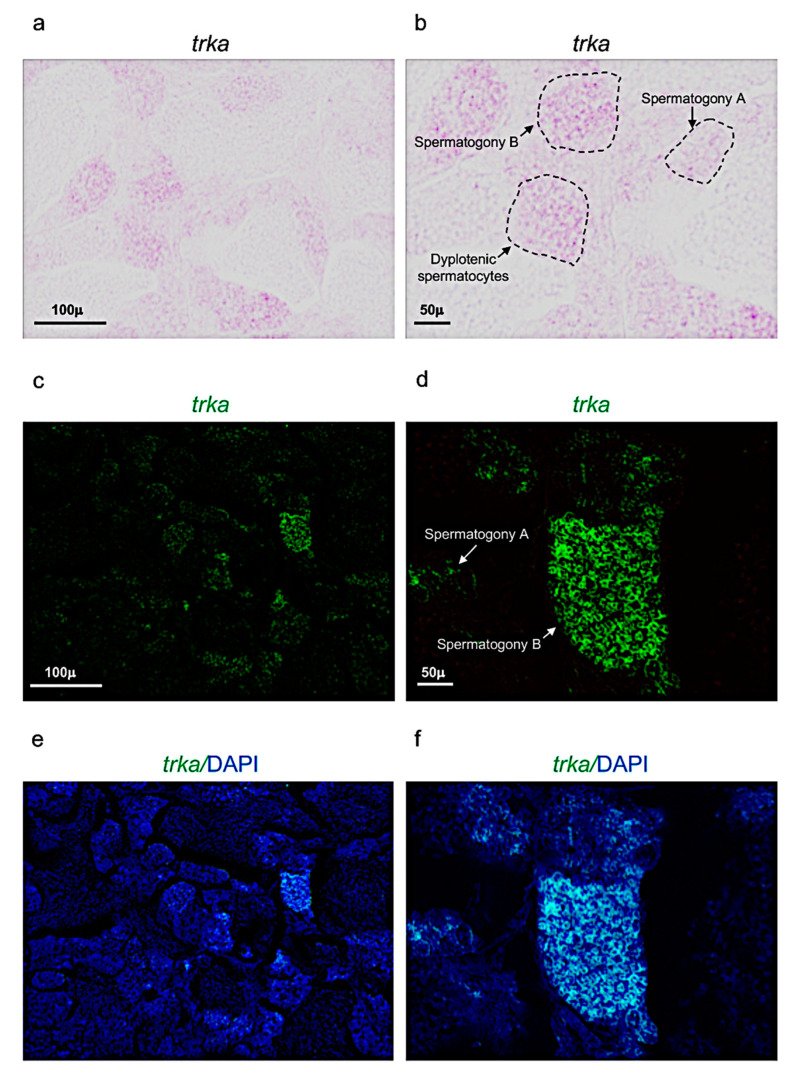
*Trka* expression pattern in adult zebrafish testis. (**a**,**b**) Chromogenic in situ hybridization of *trka* in adult zebrafish testis. (**c**,**d**) Fluorescence in situ hybridization of *trka* in adult zebrafish testis. (**e**,**f**) Fluorescence in situ hybridization of *trka* and cell nuclei (DAPI) in adult zebrafish testis. Scale bars: 100 µ (**a**,**c**,**e**); 50 µ (**b**,**d**,**f**).

**Figure 5 vetsci-09-00225-f005:**
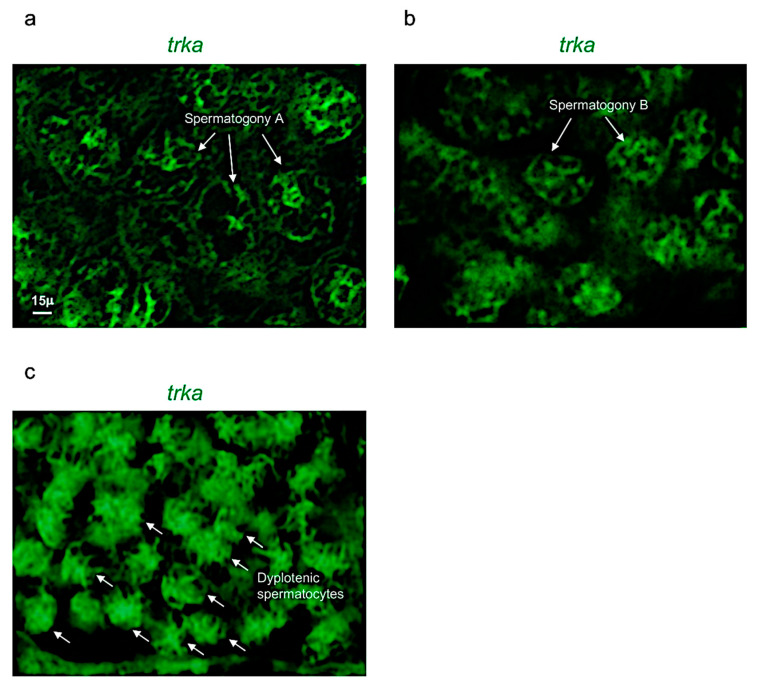
Trka is specifically expressed in spermatogony A, B and dyplotenic spermatocytes. High magnification, fluorescence in situ hybridization of *ngf*, specifically expressed in (**a**) spermatogony A; (**b**) spermatogony B; and (**c**) dyplotenic spermatocytes in adult zebrafish testis. Scale bar: 15 µ (**a**–**c**).

**Figure 6 vetsci-09-00225-f006:**
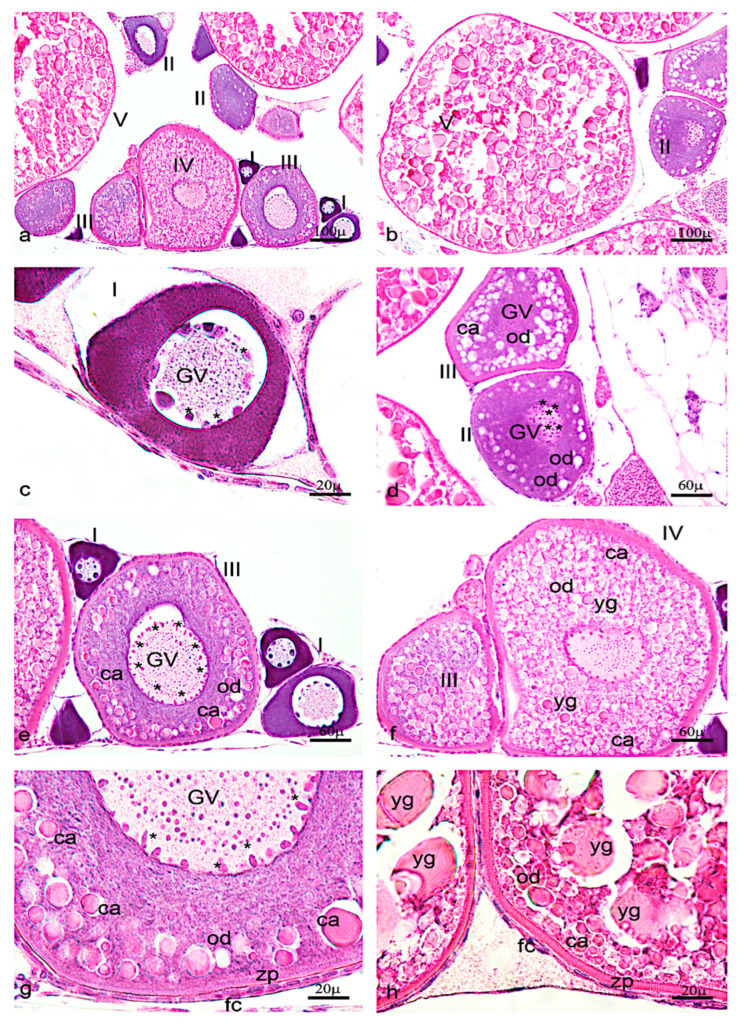
Hematoxylin-eosin staining of adult zebrafish ovary. (**a**) Overview of adult zebrafish ovary by hematoxylin-eosin staining, five oocyte stages (I-II-II-IV-V). (**b**) Oocyte stages II and V. (**c**) Oogonia characterized by large euchromatic germinal vescicle (GV), and several nucleoli peripherically located. (**d**) Oocyte during primary growth (stages II–III), characterized by an increase of nucleoli in GV, and in ooplasma were present oil droplets (od) around the GV. (**e**) Oocyte (stages III) characterized by numerous nucleoli at the periphery of GV. (**f**) numerous oil droplets and cortical alveoli (ca). (**g**) The oocyte (stages IV) is enveloped by zona pellucida (zp) and a single layer of follicular cells (fc). (**h**) Oocyte (stage V) characterized by a significant increase of number and size of the yolk globules (yg). Significant increase in the thickness of the zona pellucida and appearance of thecal layer. Scale bars are: 100 µ (**a**,**b**); 60 µ (**d**–**f**); 20 µ (**c**,**g**,**h**).

**Figure 7 vetsci-09-00225-f007:**
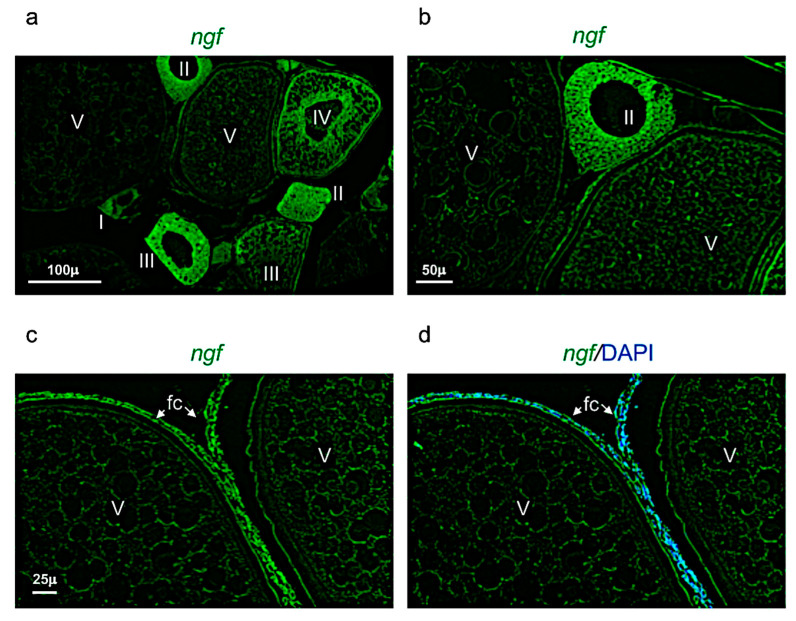
*Ngf* expression pattern in adult zebrafish ovary. (**a**) Fluorescence in situ hybridization of *ngf* in adult zebrafish ovary, mainly expressed in oocyte at different stages: II-III-IV and in follicular cells of stage V. (**b**) Fluorescence in situ hybridization of *ngf* in oocytes at II and V stages. (**c**) Fluorescence in situ hybridization of *ngf*, high magnification of follicular cells in oocyte at stage V. (**d**) Fluorescence in situ hybridization of *ngf* and cell nuclei (DAPI)*,* high magnification of follicular cells in oocyte at stage V. Scale bars: 100 µ (**a**); 50 µ (**b**); and 25 µ (**c**,**d**).

**Figure 8 vetsci-09-00225-f008:**
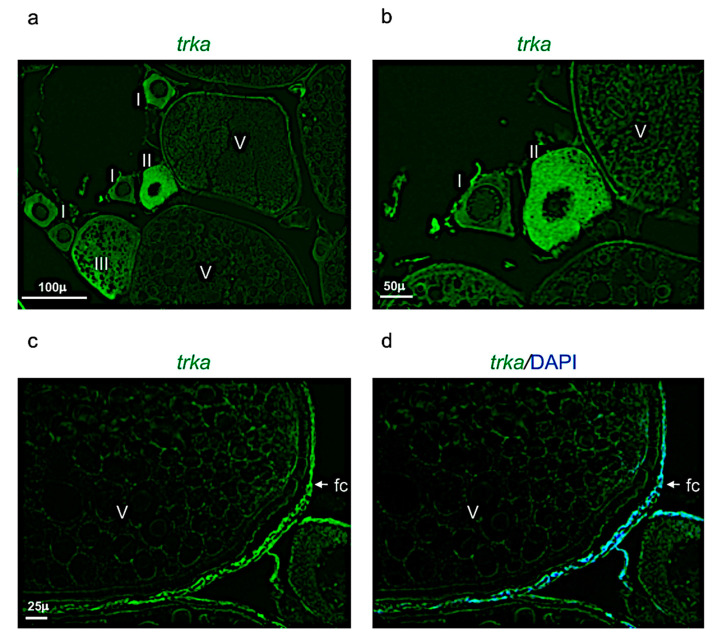
*Trka* expression pattern in adult zebrafish ovary. (**a**) Fluorescence in situ hybridization of *trka* in adult zebrafish ovary, mainly expressed in oocytes at different stages: II, III, and in follicular cells of stage V. (**b**) Fluorescence in situ hybridization of *trka* in oocytes at the stages I, II, and V. (**c**) Fluorescence in situ hybridization of *trka,* high magnification of follicular cells in oocyte at stage V. (**d**) Fluorescence in situ hybridization of *ngf* and cell nuclei (DAPI)*,* high magnification of follicular cells in oocyte at stage V. Scale bars: 100 µ (**a**); 50 µ (**b**); and 25 µ (**c**,**d**).

## Data Availability

Not applicable.

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
