# Peer review of "Expression of Nerve Growth Factor and Its Receptor TrkA in the Reproductive System of Adult Zebrafish"

_vetsci, 2022, doi:10.3390/vetsci9050225_

Round 1
Reviewer 1 Report
The author of the present manuscript examined the expression of mRNA for nerve growth factor (NGF) and the receptor through which this factor acts on the target tissue designated TrkA in the testis and ovary of adult zebrafish. The chosen methodological approach is described in sufficient detail. Although the manuscript includes results obtained using only one methodology, I think the manuscript provides interesting information. The results obtained are well documented. The discussion clearly summarizes the existing knowledge and puts it in the appropriate context to the newly acquired information. I have just few small points/comments:
- I'm not sure if the scale bares in the photos correspond to reality. Please check it.
- In lines 274 and 275: references 1 and 2 are not in appropriate format.
Author Response
Reviewer 1
In this study entitled “Expression of Nerve Growth Factor and its receptor TrkA, in the reproductive system of adult zebrafish” was examined the mRNA distribution of NGF and TrkA in the testis and ovary of adult zebrafish. It was investigated first the histology by using H/E staining and then the mRNA distribution by using chromogenic and fluorescence in situ hybridization. The results of this study suggest that mRNA of NGF and TrkA are mainly distributed in spermatogony B and spermatocytes. In the ovary of this fish, NGF and TrkA mRNAs are expressed at different stages of oocyte development. The author concluded that this neurotrophin and its receptor have an important role in the reproductive system that is conserved during vertebrate evolution.
The topic of this study is interesting, however, it can be accepted after minor revision.
The author thanks the reviewer for his comment and suggestions.
I have itemized my major concerns in the following paragraphs.
General remarks:
- It is true that the study has a single author but it is inelegant to use the first person verb in the manuscript. In my opinion, it is more appropriate to use the impersonal verb forms. In particular, in a scientific paper I suggest to use this form ….i.e. “It was observed that…., the sections were incubated, etc……”. I suggest to change these forms throughout the manuscript.
ANSWER: As suggested by the reviewer, all specific sentences have been modified.
Abstract: The conclusion is missing. Add one sentence regarding the conclusion of the study.
ANSWER: In the new version of the manuscript, a sentence has been added at the end of the abstract.
Methods:
Pag 2 line 71-72: Ethical approval number is missing
Pag 2 line 78: Specify the acronymus “PFA” (paraphormaldehyde) and check throughout the manuscript also for other .
ANSWER: The approval number has been added, and the acronymus have been checked in all manuscript.
Results :
Line131-132; line 149-150; line 177-178; line 191-192;I would avoid starting the paragraphs of the results with a sentence which is the purpose of the study. These sentences can be inserted at the end of the introduction. I suggest delete these sentences.
ANSWER: As suggested by the reviewer the sentences have been deleted, and integrated at the end of the introduction.
Discussion:
Line 225-228: This paragraph is unclear. In order to clarify the probable role of these proteins in testicular development and spermatogenesis, improve this paragraph.
ANSWER: The paragraph has been improved with more detail.
Line 235-255: The hypothesis on the functions of these protein in the ovary is missing. Please, clarify this point.
ANSWER: The paragraph has been improved in the new version of the manuscript. NGF and its receptor play a pivotal role during the differentiation of primordial ovarian follicles and the growth of the latter in secondary follicles. In detail, it has been showed that NGF/TrkA signaling pathway induced the proliferation of theca and granulosa cells, increasing the synthesis of the follicular stimulating hormone (FSH) receptor in granulosa cells. Concerning the ovulation, NGF/TrkA pathway is highly involved in this process. Indeed, this pathway, in pre-ovulatory follicle, can modulate the release of prostaglandin-E2 (PGE2), inhibiting the expression of gap-junctions and increasing the proliferation of theca cell.
Figures: Insert in the figures the negative control of in situ hybridization reactions .
Two supplementary figures 1 and 2 (ngf and trka sense probes, control in testis and ovary) have been added in supplementary material.
Reviewer 2 Report
In this study entitled “Expression of Nerve Growth Factor and its receptor TrkA, in the reproductive system of adult zebrafish” was examined the mRNA distribution of NGF and TrkA in the testis and ovary of adult zebrafish. It was investigated first the histology by using H/E staining and then the mRNA distribution by using chromogenic and fluorescence in situ hybridization. The results of this study suggest that mRNA of NGF and TrkA are mainly distributed in spermatogony B and spermatocytes. In the ovary of this fish, NGF and TrkA mRNAs are expressed at different stages of oocyte development. The author concluded that this neurotrophin and its receptor have an important role in the reproductive system that is conserved during vertebrate evolution.
The topic of this study is interesting, however, it can be accepted after minor revision.
I have itemized my major concerns in the following paragraphs.
General remarks:
- It is true that the study has a single author but it is inelegant to use the first person verb in the manuscript. In my opinion, it is more appropriate to use the impersonal verb forms. In particular, in a scientific paper I suggest to use this form ….i.e. “It was observed that…., the sections were incubated, etc……”. I suggest to change these forms throughout the manuscript.
Abstract: The conclusion is missing. Add one sentence regarding the conclusion of the study.
Methods:
Pag 2 line 71-72: Ethical approval number is missing
Pag 2 line 78: Specify the acronymus “PFA” (paraphormaldehyde) and check throughout the manuscript also for other .
Results :
Line131-132; line 149-150; line 177-178; line 191-192;I would avoid starting the paragraphs of the results with a sentence which is the purpose of the study. These sentences can be inserted at the end of the introduction. I suggest delete these sentences.
Discussion:
Line 225-228: This paragraph is unclear. In order to clarify the probable role of these proteins in testicular development and spermatogenesis, improve this paragraph.
Line 235-255: The hypothesis on the functions of these protein in the ovary is missing. Please, clarify this point.
Figures: Insert in the figures the negative control of in situ hybridization reactions .
Author Response
Reviewer 2
The author of the present manuscript examined the expression of mRNA for nerve growth factor (NGF) and the receptor through which this factor acts on the target tissue designated TrkA in the testis and ovary of adult zebrafish. The chosen methodological approach is described in sufficient detail. Although the manuscript includes results obtained using only one methodology, I think the manuscript provides interesting information. The results obtained are well documented. The discussion clearly summarizes the existing knowledge and puts it in the appropriate context to the newly acquired information. I have just few small points/comments:
ANSWER: The author thanks the reviewer for his comments.
- I'm not sure if the scale bares in the photos correspond to reality. Please check it.
ANSWER: The scale bars have been checked in the previous and new versions of the manuscript, according to microscope calibration published recently (Cacialli P., et al., Nature Communications 2021).
- In lines 274 and 275: references 1 and 2 are not in appropriate format.
ANSWER: Concerning the format of the references, all of them have been imported using EndnoteX9, as recommended by the ACS style guide and reported in the Instructions for Authors of the journal. .